# Microstructural Properties of Particle-Reinforced Multilayer Systems of 316L and 430L Alloys on Gray Cast Iron

Mohammad Masafi [1,*], Heinz Palkowski [1] and Hadi Mozaffari-Jovein [2]

1   Institute of Metallurgy, Clausthal University of Technology, Robert-Koch-Strasse 42,
    38678 Clausthal-Zellerfeld, Germany; heinz.palkowski@tu-clausthal.de
2   Institute of Materials Science and Engineering Tuttlingen (IWAT), Furtwangen University, Kronenstraße 16,
    78532 Tuttlingen, Germany; hadi.mozaffarijovein@hs-furtwangen.de
*   Correspondence: mohammad.masafi.1@tu-clausthal.de

**Abstract:** Gray cast iron (GJL) is known for its excellent damping property and high thermal conductivity, thanks to its unique lamellar graphite and pearlite structure. In a recent study, laser metal deposition (LMD) was explored as a potential process to enhance the corrosion resistance and wear mechanism of this tribological system. The focus was on laser cladding of gray cast iron using two different of stainless-steel materials, namely 430L and 316L, combined with TiC and WC particles. To create the samples, a multilayer coating system was employed. A comparative analysis of the microstructures was performed to understand the interaction of the laser beam with the material (composite materials). Surface properties were then characterized using light microscopy and electron microscopy (SEM) before and after subjecting the samples to a shock corrosion test, simulating automotive conditions. Additionally, phase analyses were performed at the interfaces between the coatings and the substrate, with particular attention given to the behavior of the graphite lamellae at these interfaces. This study aims to provide valuable insights into the potential improvements that can be achieved through laser cladding on gray cast iron, specifically in terms of corrosion resistance and wear mechanisms. By analyzing the microstructures and surface properties, researchers can gain a better understanding of the performance and durability of the coated samples.

**Keywords:** gray cast iron; brake disc; emission particle; graphite phase; laser metal deposition (LMD); laser cladding; stainless-steel coating; multilayer coating; sigma phase

## 1. Introduction

Lamellar gray cast iron (GJL, type A) with a pearlite matrix is known for its excellent vibration-damping properties, good heat capacity, and thermal conductivity. These characteristics make it a desirable material for various applications. Its good castability and machinability further contribute to its acceptance. One of the main reasons for its wide usage is its cost-effectiveness. Its low production cost and high availability make it a preferred choice in the European market. This affordability, combined with its desirable properties, makes it suitable to produce thin-walled and complex castings. For example, it is commonly used in the manufacturing of gear boxes and crank cases, brake discs, pumps, and valves. Specifically, for brake discs, GJL alloys are favored due to their thermal conductivity and eigenfrequency, which ensure efficient heat dissipation and optimal braking system performance [1].

However, braking systems are not only responsible for stopping vehicles but also for the direct emission of particulate matter into the air. Continued usage and friction between the brake components lead to wear, resulting in the generation of particulate matter. These emitted particles primarily belong to the size classes of fine dust particles, with a diameter ≤ 10 μm. This emission occurs due to the wear and tear on brake pads and discs, which release a significant amount of particulate matter into the environment. In

fact, studies have shown that brake wear contributes to 55% of total traffic-related PM10 emissions (excluding exhaust gases) and 21% of PM2.5 emissions [2]. This distinction makes brake wear particulate matter particularly relevant to human health, prompting extensive scientific investigations [3]. To mitigate the impact of these emissions on human health, efforts have been focused on improving the surface of brake disks. Although there are currently no legislative limits in place for brake wear particle emissions, it is imperative to limit their production [4].

Recent academic studies have highlighted the potential of reducing particulate matter emissions by enhancing the wear resistance of brake surfaces through heat treatments [5] or the application of wear-resistant coatings [6–9]. Coating has emerged as an efficient and effective method to address the issue of particulate matter emissions from braking systems. One of the primary advantages of coating is its ability to reduce emissions directly and indirectly by enhancing wear and abrasion mechanisms. Additionally, the use of wear-resistant coatings has attracted significant attention in recent years. These coatings, typically composed of materials such as ceramics, polymers, or composites, are applied to the surface of brake components to provide a protective barrier against wear and friction. Furthermore, research efforts have focused on developing advanced coating processes that offer superior wear resistance and durability. Techniques such as physical vapor deposition (PVD), chemical vapor deposition (CVD), and thermal spray have been explored to deposit high-performance coatings onto brake surfaces. In conclusion, coating has emerged as a promising solution to mitigate particulate matter emissions from braking systems.

After careful consideration and evaluation of various coating methods, LMD has been identified as a suitable technique for coating the surface of GJL. Laser cladding is an advanced surface finishing process that offers several advantages over conventional methods. It can produce high-performance coatings that are metallurgically bonded and fully dense on metallic substrates [10]. This makes laser cladding a suitable deposition technique for welding on GJL brake arches [11]. The process of laser cladding involves using a laser beam to melt and fuse a powdered coating material onto the surface of the substrate. Additionally, laser cladding allows for precise control over the thickness and composition of the coating. Furthermore, laser cladding has been recognized as an important material processing technique and a form of reprocessing technology [12]. LMD offers several advantages, including the ability to create a strong diffusion and bond between the coating and GJL.

The use of a multilayer system was adopted in the current research due to the consideration of seasonal and environmental factors. The selection of coating materials is based on their corrosion resistance, wear resistance, and compatibility with GJL. It is worth noting that the development and optimization of coating materials and multilayer systems are ongoing areas of research. Continued research and development in this field will further enhance the performance and environmental impact of braking systems, ultimately contributing to a cleaner and safer transportation environment.

To evaluate the quality of a multilayer system using laser metal deposition (LMD), a comprehensive analysis is required. This study aims to characterize the coating, interfaces, and behavior of the graphite phase during laser cladding. The starting materials for this analysis are cast iron samples coated with two types of steel powders and two types of hard particles. The characterization involves metallographic and microscopic examination of the coating and interfaces. Metallography provides insights into the microstructure of the coating, including the distribution of different phases and any potential defects, such as cracks or voids. Microscopic analysis allows for a closer examination of the coating and interfaces, providing information on the bonding between layers and the behavior of the graphite phase. By conducting this analysis, it will be possible to identify any quality deficiencies in the multilayer system. This information will be crucial for optimizing the coating process and ensuring the desired corrosion resistance of brake disks. Additionally, it will help in understanding the behavior of the graphite phase, which is important for the overall performance and durability of the coating.

## 2. Materials and Methods

Gray cast iron (GJL 150) with graphite A according to EN 1561, as seen in Figure 1a, has a degree of saturation corresponding to the eutectic point (Figure 1b). The chemical composition of this GJL was analyzed by optical emission spectrometry (OES), as shown in Table 1. The ratio between the total carbon content of the melt and the carbon content of the eutectic composition according to Eq. 1 is Sc 0.98. Here, the influence of the accompanying elements on the shift of the eutectic point (eutectic, eutectic temperature) is considered. Thus, a saturation value of Sc 0.98 indicates a hypoeutectic cast iron.

$$Sc = \%C/(4.26 - 1/3(Si\% + P\%)) \tag{1}$$

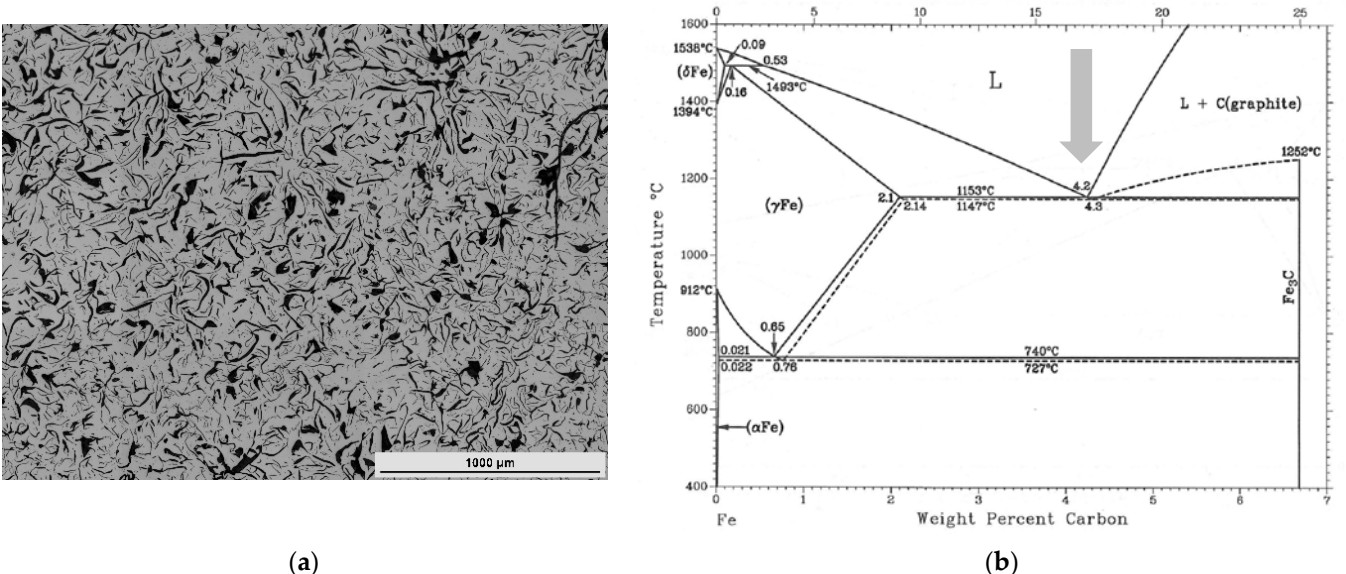

(**a**)                                                                 (**b**)

**Figure 1.** Cast iron with lamellar graphite. (**a**) Form of lamellar graphite. (**b**) Iron–carbon phase diagram and eutectic point (adapted) [13].

**Table 1.** Chemical composition of the powder and substrate.

| Element (wt.%) | GJL 150 | 316L | 430L |
|:---:|:---:|:---:|:---:|
| C | $3.50 \pm 0.1$ | 0.01 | 0.001 |
| Si | $2.00 \pm 0.1$ | 0.8 | 0.68 |
| Mn | $0.60 \pm 0.05$ | 1.5 | 0.09 |
| P | $<0.10 \pm 0.02$ | - | 0.01 |
| S | $<0.08 \pm 0.02$ | <0.01 | <0.01 |
| Cu | $0.20 \pm 0.02$ | 0.0 | 0.0 |
| Cr | $0.20 \pm 0.02$ | 17.00 | 16.44 |
| Mo | $0.35 \pm 0.1$ | 2.50 | 0.075 |
| Ni | <0.20 | 12.00 | <0.60 |
| Sn | <0.10 | - | - |
| N | - | - | - |
| Fe | Balance | Balance | Balance |

Table 1 shows the chemical composition of the input materials used in this project, such as GJL and steel powders 316L and 430L. The chemical composition of the gas-atomized powders used to coat the samples is reported by the supplier, Höganäs. Two different stainless-steel powders (ferritic stainless steel 430L and austenitic stainless steel 316L) were used as matrix coating material, and TiC and WC were used as hard particles for coating. The size of the matrix particles was determined to be 20–53 μm (max. 5% oversize and undersize) for 316L, 430L, and WC. The normal grain fraction of TiC is in the range of 45–100 μm (max. 10% oversize and undersize).

The laser metal deposition process is described as follows. There are two containers. The first one is filled with matrix powder: either steel powder 316L or 430L. The other is filled with TiC or WC hard particles. The powders are mixed in different proportions in the equipment and heated under the action of laser or the steel powder is melted and applied to the substrate. A complete deposition welding process is performed under an inert gas such as argon in an isolated and closed cabinet.

Two coating layers are deposited on the GJL substrate as described in Figure 2a. The first is an adhesion promoter and serves to improve the corrosion resistance of the substrate, as it is austenitic stainless steel. This layer has better mechanical properties and thermomechanical properties. To reduce wear and emission particles, the second layer is coated on top of the first layer. Due to the hardness of the particles, the second layer can be used for wear reduction. After coating, the surface is smoothed, and the roughness is reduced by a grinding process. The roughness of the coating surface after the grinding was between 3 until 6 μm.

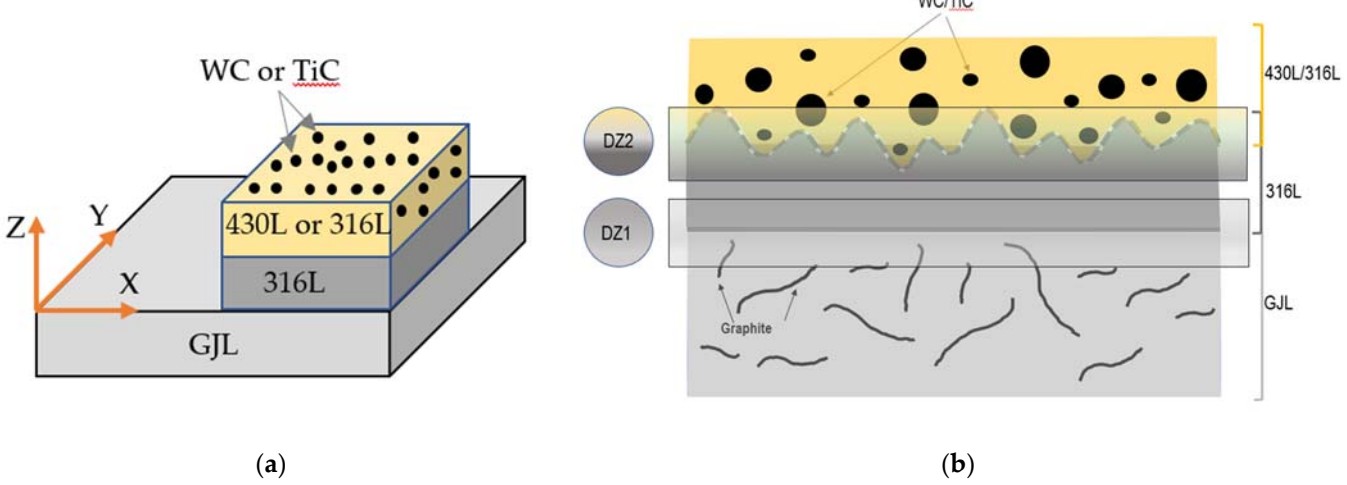

(**a**)          (**b**)

**Figure 2.** Schematic 3D and 2D representation of the manufacturing of the samples. (**a**) Two coatings are deposited on the substrate: 316L followed by 316L or 430L with hard particles WC/TiC; (**b**) two diffusion zones (DZ1 and DZ2).

There are two diffusion zones—DZ1 and DZ2—as shown in Figure 2b. In our research, these two diffusion zones were characterized as interfaces. In DZ1 and DZ2, the phases present their transformations, and material defects were also detected. For each combination of coating, three samples were processed and used for investigation.

The first type of sample was a combination of 316L (as the first layer) and 430L with WC as a hard particle as the top layer. The second type was comprised 316L for both layers and TiC as hard particles. The third consisted of 316L for both layers and WC as hard particles. All three combinations are listed in Table 2.

**Table 2.** List of investigated combinations.

|  | Combination 1 | Combination 2 | Combination 3 |
|---|---|---|---|
| Substrate | GJL | GJL | GJL |
| First layer | 316L | 316L | 316L |
| Top layer | 430L | 316L | 316L |
| Hard particle | Spherical WC | Polygonal TiC | Spherical WC |

The LMD system was used for this purpose. The process parameters used in this study are listed in Table 3.

**Table 3.** Parameters used to manufacture the samples.

| Parameter | Combination 1 | Combination 2 | Combination 3 |
| --- | --- | --- | --- |
| Laser power | <10 kW | <10 kW | <10 kW |
| Total multilayer thickness | <400 μm | <400 μm | <400 μm |
| Thickness of first layer | 100–150 μm | 100–150 μm | 100–150 μm |
| Thickness of top layer | 200–250 μm | 200–250 μm | 200–250 μm |
| Size of WC | 20–53 μm | - | 20–53 μm |
| Size of TiC | - | 45–100 μm | - |
| Hard particle distribution in top layer | 20% | 40% | 20% |

The samples were subjected to a shock corrosion test according to an automotive test specification. The test comprised 1200 braking cycles, a 20 h salt spray test according to DIN EN ISO 9227 [14], 16 h humid heat storage according to DIN EN ISO 6270-2 [15], and 20 h storage in a normal climate (RT).

The microstructure was determined via metallographic investigation. The samples were treated with a V2A etchant (50% HCl, 10% $HNO_3$, and 40% $H_2O$) at 50 °C with an etching time of 120 s and with nitric acid (3% $HNO_3$, 97% $C_2H_5OH$) at room temperature with an etching time of 20 s. Light optical microscopic (LOM) and scanning electron microscopic (SEM) examinations were used to characterize the coating. The focus of the microscopic examination was on analyzing wear, cracking, corrosion, and delamination. The corrosion test procedures and evaluation of the corrosion results are based on test specification F.350.016 of the automotive industry and are not explicitly considered here. One side is necessary for good braking at high speed and high friction in a short time, and the other side must exhibit reduced dust emission. Hard particles have very high hardness, but the matrix of the coating has close to the same hardness as substrate or a slightly more. The matrix of the top-layer coating helps to support and to hold the hard particles, while the brakes generate enough friction and produce less dust emission.

## 3. Results and Discussion

After the samples had passed the shock corrosion test, they were analyzed. Figure 3 shows the surfaces of the specimens for all three conditions of preparation after and before the corrosion test. The surface of the coating before the shock corrosion test is shown in Figure 3a. Because the surfaces of the various coatings before the test are similar, just an illustration of the surfaces is shown here. The surfaces of the different multilayer systems after the test are visualized in in Figure 3b–d. All surfaces exhibited a crack network and delamination after the corrosion test. Figure 3b,d show the crack network and delamination. The delamination, the depth of the cracks, and the crack network in Figure 3b are larger than in the other Figures. Here, the cracks are more open than in Figure 3d. In Figure 3d, there is already delamination, but the cracks are not more open. The cracks in Figure 3d look like they are coming from the substrate, perhaps a weak area of GJL. Further detailed analysis is presented in other sections. Here, the focus is on the macroscopic description of surfaces.

Cross sections of the samples are shown in Figure 4. Figure 4a shows an example of the surface condition after coating and grinding the 316L-316L multilayer system with WC so that it is easier to imagine the condition of the surface after coating and grinding, as it was before the test, and how the microcracks and cracks developed during the test.

Galvanic corrosion is a typical type of corrosion in multilayer systems. Figure 4c shows that when the cracks arrive at the substrate (GJL), corrosion occurs more quickly due to galvanic corrosion between the high negative electrochemical potential of GJL (−0.6 to −0.8 V) and the 316L layer (0.0 until −0.2 V) [16]. Galvanic beam speed depends on

the difference between the energy of active and noble elements. The greater the difference, the faster corrosion is performed, since the difference rate between the galvanic energy of 316L and that of GJL is greater than the difference between the galvanic energy of the 430L layer and that of the 316L layer. Therefore, it can be reasoned that when cracks reach the GJL, further corrosion occurs more rapidly.

The graphite in DZ1 was exposed during laser beam irradiation. Different thermal expansions of the materials cause this phenomenon. Depending on the graphite form and laser and coating parameters, this phenomenon quantitatively varies. Different morphologies of hard particles and different diffusion qualities lead to different wear. For example, closely placed TiC particles and the creation of intermetallic phases fostered cracking in the multilayer system with TiC. On the other hand, a combination of 316L with 430L simplified the formation of brittle phases during braking. Therefore, the 316L multilayer system has both a layer with WC hard particles and better quality in comparison with other samples in this study.

Figure 5 shows the effect of laser treatment in DZ1 for combination 2. The graphite is influenced by the laser beam, which later leads to the formation of cracks. This phenomenon was observed in all samples. After etching, decarburization of the substrate was detected, as shown in Figure 5b. Where a graphite network existed, the graphite was compressed during laser beam processing. This is probably due to the difference in thermal expansion behavior between the matrix and the graphite lamellar and is a reason for the pressing or squeezing of the graphite. Due to the partially tightly arranged sharp-edged TiC particles, the formation of long cracks was heightened, as can be seen in Figure 5c,d.

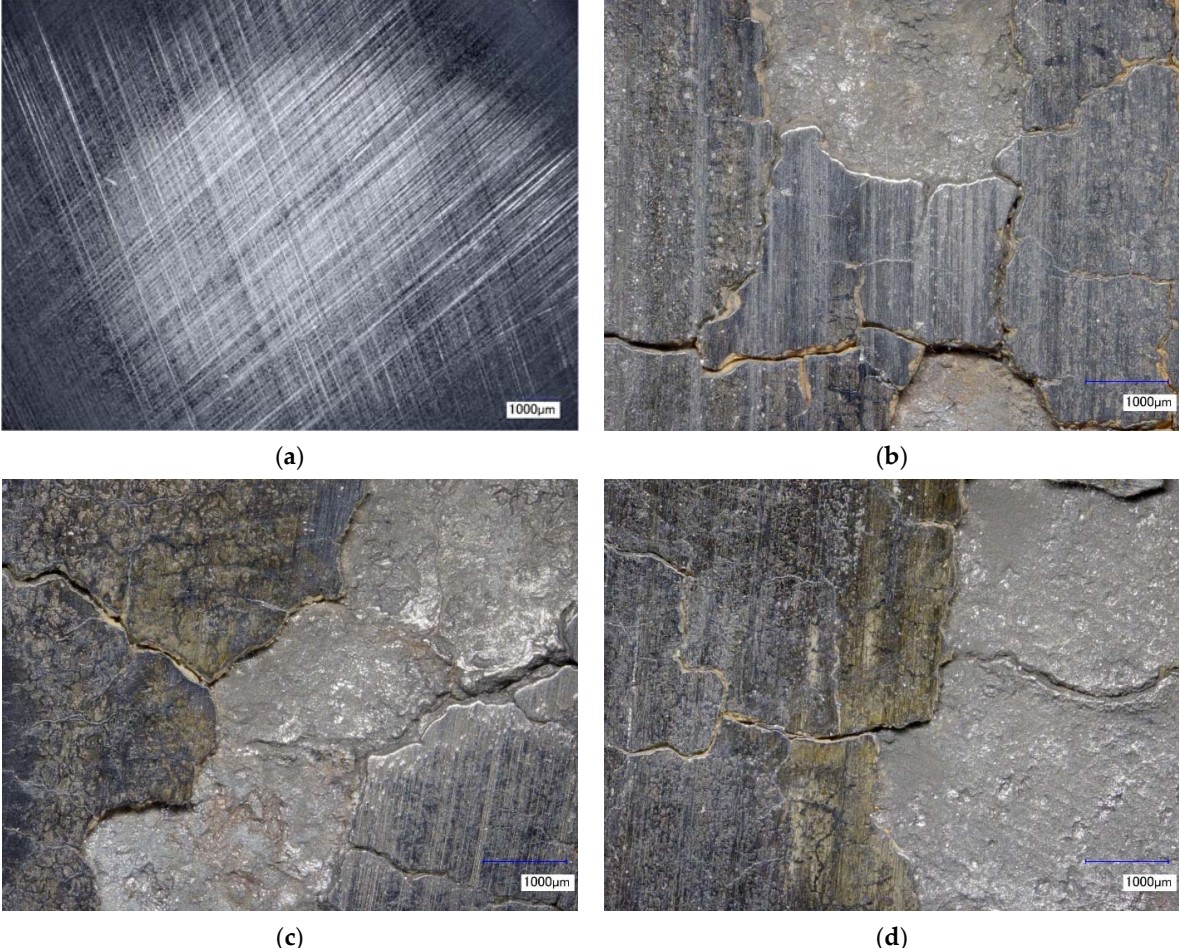

(**a**)　　　　　　　　　　　　　　　(**b**)

(**c**)　　　　　　　　　　　　　　　(**d**)

**Figure 3.** Surfaces of the samples before (**a**) and after (**b**–**d**) the shock corrosion tests: (**b**) 316L-430L with WC; (**c**) 316L-316L with TiC; (**d**) 316L-316L with WC. A crack network and delamination were detected macroscopically in all the samples after corrosion tests.

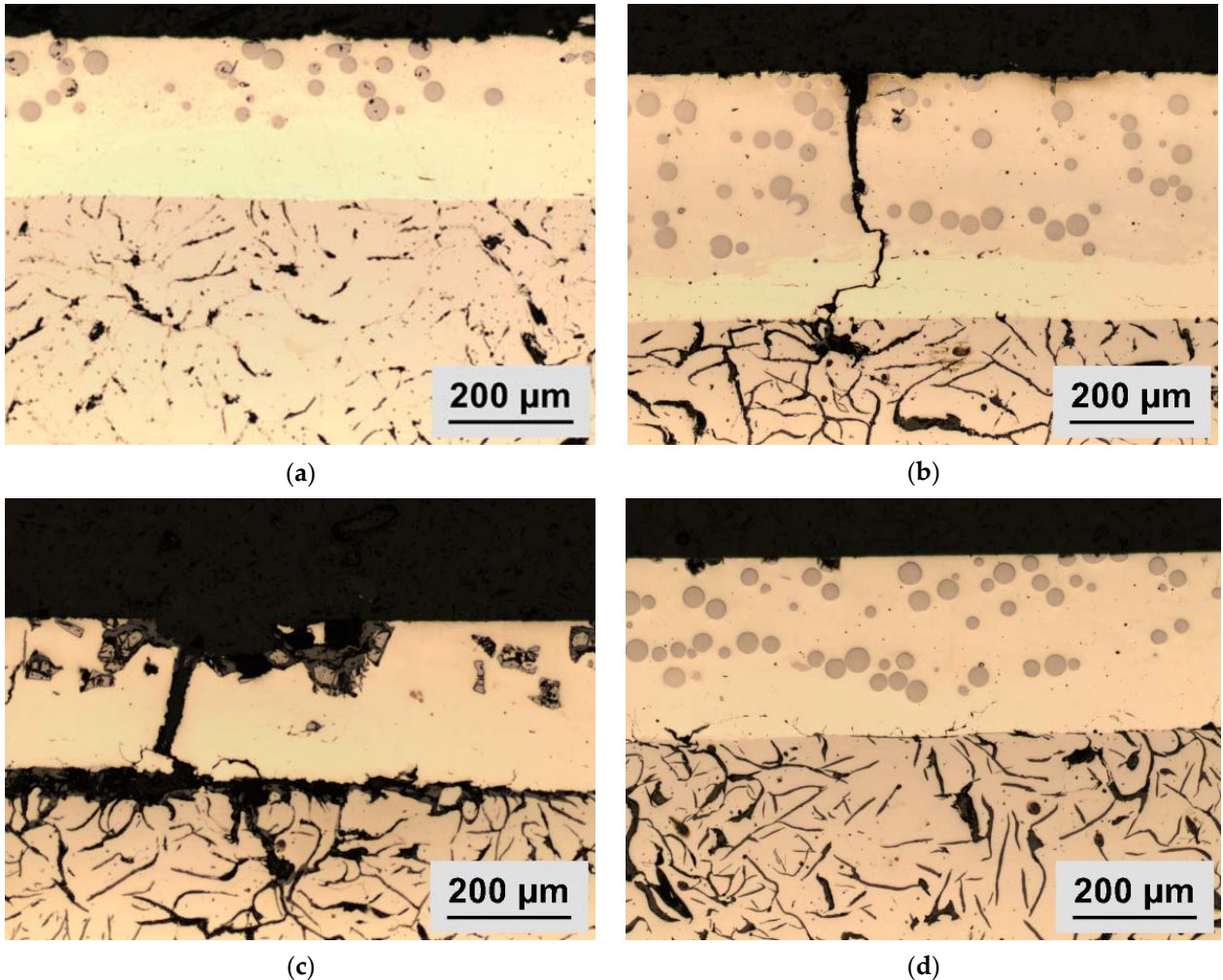

**Figure 4.** Cross sections of samples of different combinations before ((**a**) exemplar for 316L-316L with WC) and after (**b**–**d**) the shock corrosion tests: (**b**) 316L-430L with WC; (**c**) 316L-316L with TiC; (**d**) 316L-316L with WC.

Figure 6 shows metallographic images of the investigated samples around the DZ2 diffusion zone (transition between the two coatings) after V2A pickling. In this area, an unusual phase was detected only for samples under condition 1, namely the 316L and 430L layers. This is the sigma phase.

Figure 7 shows a top-layer analysis of the multilayer system of sample 2, which was coated with 316L as both layers and TiC as hard particles. The phases were formed in this layer during diffusions and transformations. It is very important to analyze this zone with SEM to better understand why the cracks were formed more easily. Not only does this process depend on closely deposited TiC hard particles and their morphology, but the phases also have an important role in the formation of cracks. SEM analysis shows that the TiC dissolved and diffused into the matrix material (316L) during the coating process (Figure 7a). It can be seen that the matrix material consists of two phases, namely a Ni-containing austenite phase and a Cr-containing ferrite phase. This phase separation occurs due to the rapid solidification of the melt during the coating process. Figure 7c exhibits the points of measurement considered in EDX analyses, and Table 4 lists the atom concentrations. Table 4 also shows the chemical composition measured with EDX at each measuring point near a TiC hard particle. There is Ti in the S1, S2, and S3 measuring points, which means that Ti was dissolved from TiC and diffused in the 316L matrix. This measurement proves that Ti diffuses from TiC. The formed intermetallic phase is most likely TiC in iron-containing matrix materials [17,18].

**Table 4.** Analysis of the chemical composition of the samples according to Figure 7c in atomic percent.

| Spectrum | C (%) | Ti (%) | Cr (%) | Fe (%) | Ni (%) | Mo (%) |
|---|---|---|---|---|---|---|
| S 1 | 1.77 | 8.38 | 16.10 | 61.77 | 10.68 | 1.29 |
| S 2 | 1.11 | 3.69 | 16.76 | 66.86 | 10.67 | 0.93 |
| S 3 | 1.70 | 5.71 | 18.86 | 61.64 | 10.67 | 1.27 |
| S 4 | 1.23 | 3.37 | 16.77 | 66.34 | 11.29 | 0.98 |
| S 5 | 2.26 | 7.45 | 17.89 | 60.39 | 10.71 | 11.96 |
| S 6 | 1.35 | 5.01 | 16.96 | 64.68 | 11.13 | 0.87 |
| S 7 | 1.63 | 7.59 | 16.98 | 62.20 | 10.39 | 1.21 |
| S 8 | 1.20 | 5.65 | 17.21 | 63.84 | 11.02 | 1.09 |
| S 9 | 1.48 | 5.64 | 17.81 | 62.62 | 11.35 | 1.10 |

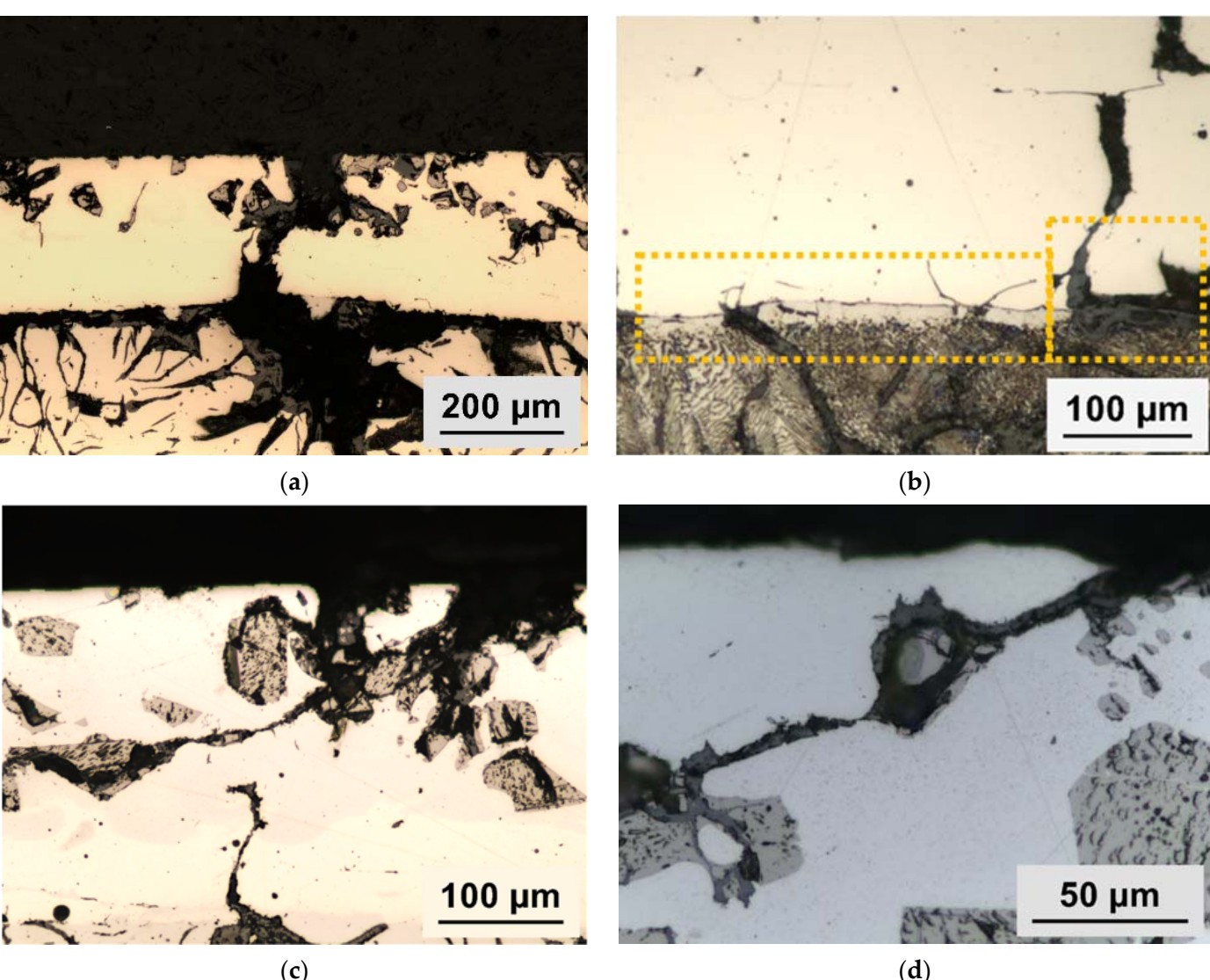

**Figure 5.** Cross section of the 316-316 interface with TiC and the behavior of graphite and the substrate due to thermal expansion during laser beam processing: (**a**) exposed graphite; (**b**) decarburization in the left yellow box and crushed graphite in the right yellow box; (**c**,**d**) agglomeration of TiC particles with crack initiation.

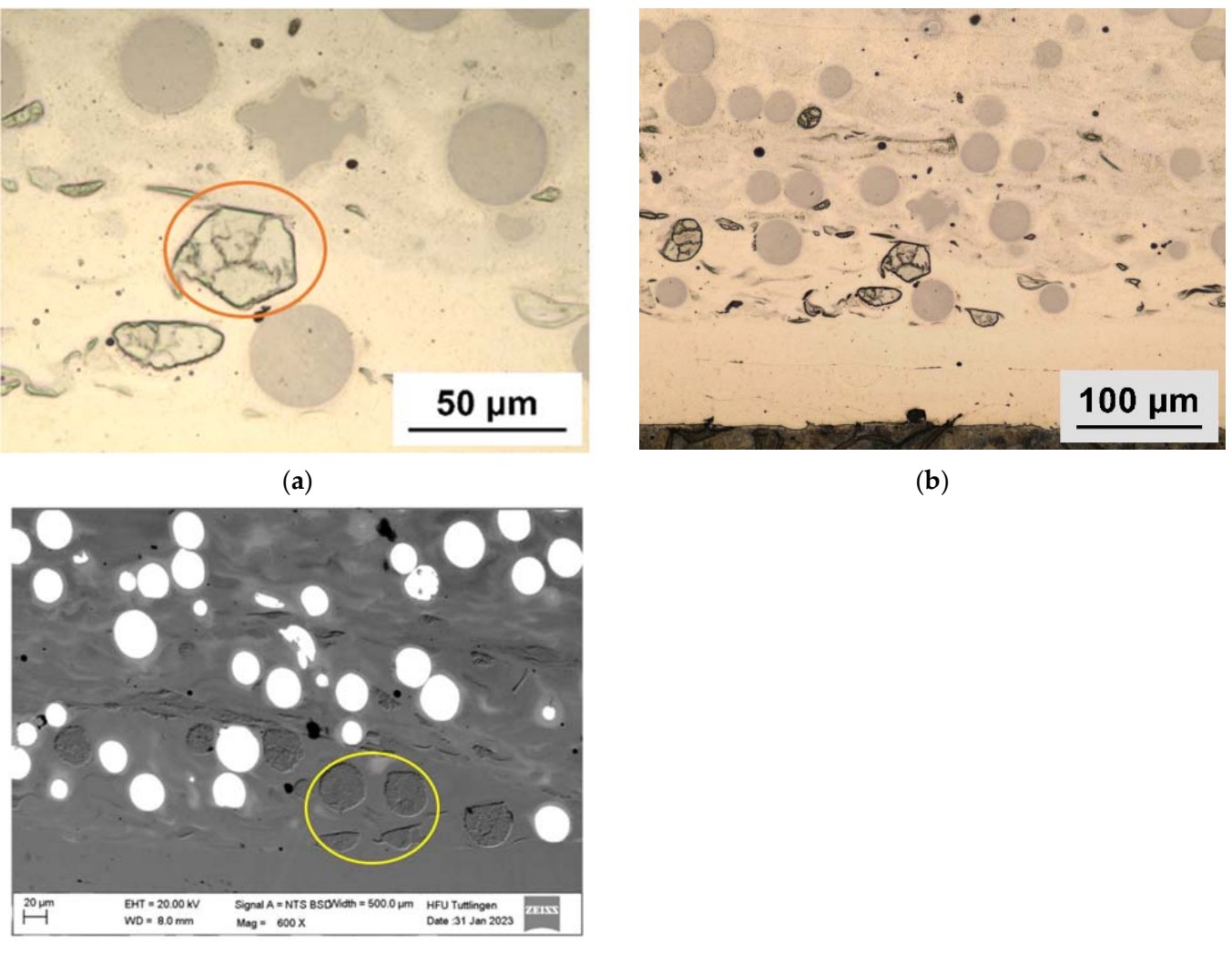

**Figure 6.** Detection of the sigma phase: (**a**) cross section of 316L-430L with WC after V2A pickling and the sigma phase in orange circle; (**b**) LOM image and position of sigma phase in the DZ2; (**c**) SEM sigma phase in yellow circle.

Further SEM and light microscopy studies showed that transcrystalline fracture through the carbide crystal the WC-containing coatings led to the dissolution of the WC into the matrix material (Figure 8a). The reaction of the alloy constituents with the WC particles led to a lowering of the melting temperature of the WC particles and, thus, to the formation of intermetallic phases containing tungsten during resolidification. Furthermore, it can be seen that cracks formed by the thermal stresses run unhindered through the WC particles. A new phase formed on the surface of the WC particles (Figure 8b). In this area, the resistance of the material to crack propagation decreases. The crack runs directly through the WC hard particle (Figure 8a). Considering the possible phase states of the ternary phase diagram, as seen in Figure 9, it is most likely an intermetallic phase ($\tau_1$). Table 5 shows the chemical composition measured with EDX at each measuring point near a broken WC hard particle. This measurement proves how the W of WC diffuses out. At measuring points S4, S5, and S6, an amount of W is also present.

A thermodynamic phase reaction occurred here. This means that the chemical composition of the microstructure does not correspond to the chemical composition of the raw alloy.

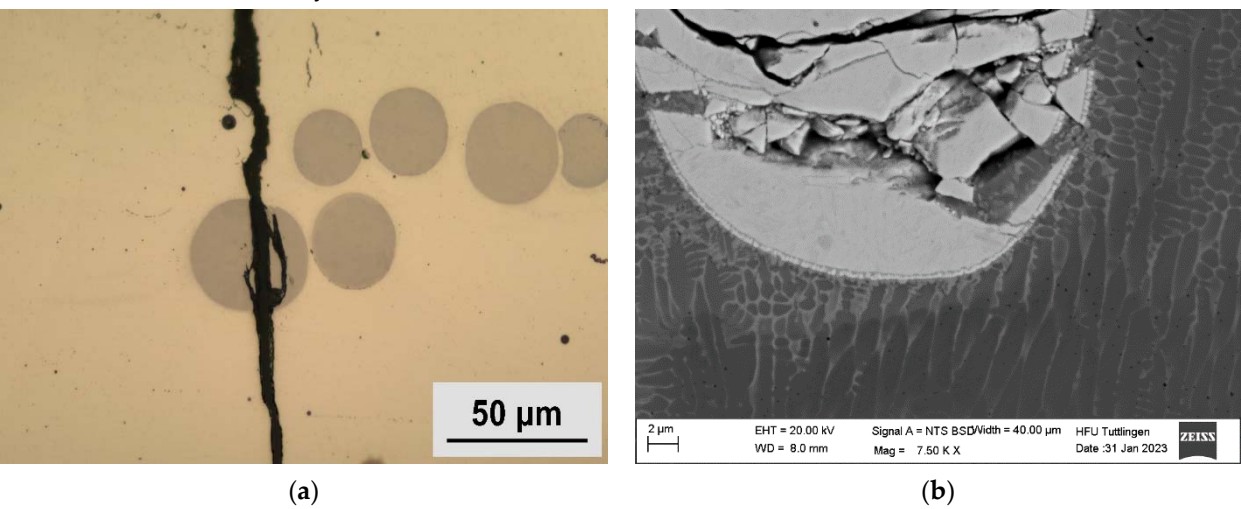

**Figure 7.** SEM cross-section images of the 316L-316L sample coating and the hard TiC particles in the top layer: (**a**) formation of a Ti-containing intermetallic phase in the 316L matrix; (**b**) phase decay in the matrix material (light phase: austenite; dark phase: ferrite); (**c**) measuring points of EDX analysis.

**Figure 8.** SEM analysis of the third sample with 316L-316L coating and WC hard particle: (**a**) crack propagation in the WC hard particle; (**b**) fractured WC in the matrix.

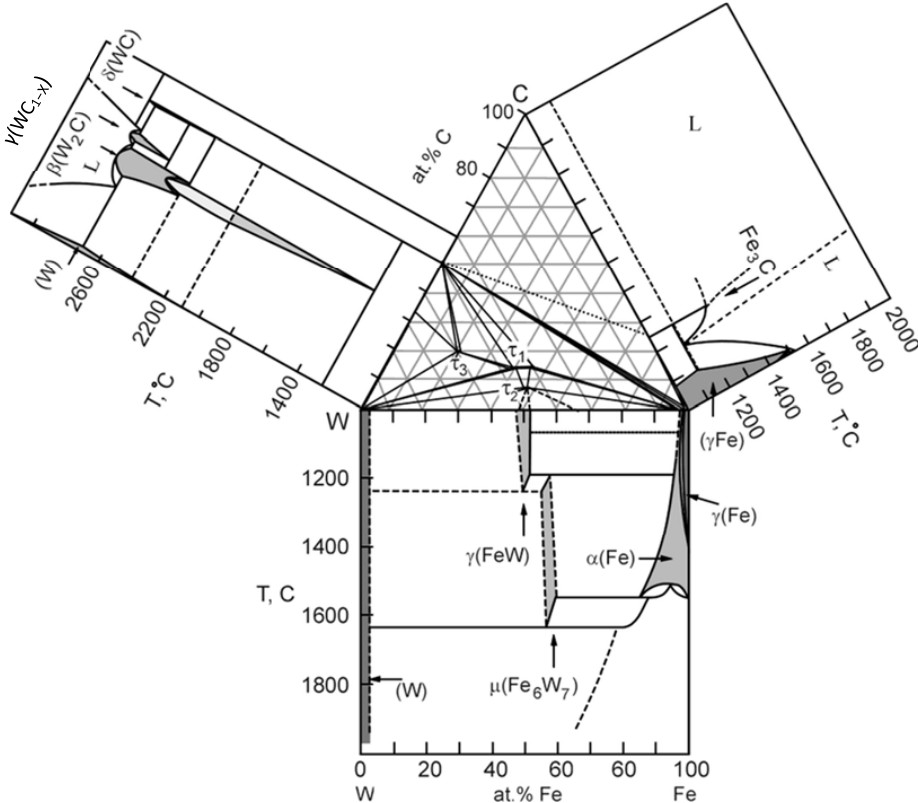

**Figure 9.** Isothermal section in the Fe-W-C ternary system at 1000 °C (reprinted) [19].

**Table 5.** Analysis of the chemical composition of the samples according to Figure 10 in atomic percent.

| Spectrum | C (%) | Cr (%) | Fe (%) | Ni (%) | W (%) |
|----------|-------|--------|--------|--------|-------|
| S1 | 12.42 | 10.45 | 34.06 | 4.66 | 38.40 |
| S2 | 11.83 | 10.73 | 31.45 | 3.64 | 42.35 |
| S3 | 4.68 | 13.96 | 46.63 | 6.87 | 27.85 |
| S4 | 4.27 | 17.35 | 59.09 | 9.21 | 10.09 |
| S5 | 4.29 | 19.79 | 55.49 | 8.47 | 11.96 |
| S6 | 2.84 | 18.58 | 59.68 | 9.84 | 9.05 |
| S7 | 2.26 | 16.61 | 66.78 | 11.89 | 2.47 |
| S8 | - | 16.70 | 68.53 | 12.03 | 2.73 |
| S9 | 2.62 | 17.28 | 63.67 | 10.67 | 5.75 |

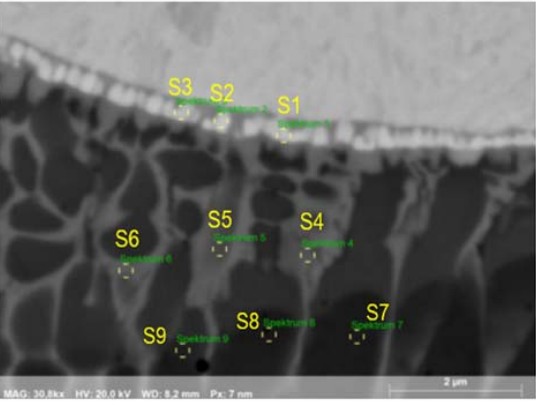

**Figure 10.** Selected positions for EDX analysis of sample 3 in the area of WC particles corresponding to the data presented in Table 5.

## 4. Conclusions

A multilayer coating system with laser deposition welding on cast iron was characterized. The thermodynamic behavior of TiC, WC hard particles with matrix powder, and possible defects after an automotive brake corrosion test were analyzed. The quality of the coating system depends on the following conditions: the morphology and distribution of the hard material particles [20,21], the formation of a brittle sigma phase, the tendency of the composite to form intermetallic phases, adhesion of the wear protection layer in the multilayer system, and the thermal behavior of the components involved.

When the cracks arrive at the substrate (GJL), corrosion occurs more quickly due to galvanic corrosion between the substrate and coating layers, thus negatively affecting the corrosion protection and resulting in spalling of the layer (see Figure 4).

The melting and solidification conditions lead to the formation of local thermomechanical stresses that destroy the graphite lamellae in the cast iron in the diffusion zone between GJL and first 316L coating layer (DZ1) (see Figures 4c and 5a).

Decarburization can be observed near the substrate surface in DZ1, leading to a decrease in the strength of the substrate (see Figure 5b).

The inhomogeneous distribution of hard particles and the non-uniform of morphology of TiC have a considerable influence on the mechanical properties of the coating material, which are negatively affected, resulting in cracking behavior, which, in turn, strongly influences the wear behavior of the multilayer coating [22,23] (see Figure 5c,d).

Material investigations confirmed the presence of sigma phases in DZ2, which were metallographically visible after pickling with "V2A pickle" (see Figure 6).

Following diffusion of Ti into the 316L matrix, the matrix material consisted of two phases—austenite and ferrite—with different compositions (see Figure 7).

The melting temperature of the WC particles can be reduced by the diffusion of the alloying constituents from 316L powder into the WC hard material, leading to the formation of new τ1-intermetallic phases (see Figures 8–10).

**Author Contributions:** This research article is written by several authors with the following details. Conceptualisation: M.M.; Methodology, M.M., H.P. and H.M.-J.; Validation, M.M., H.P. and H.M.-J.; Formal analysis, M.M.; Investigation, M.M.; Resources, M.M., H.P. and H.M.-J.; Data maintenance, M.M., H.P. and H.M.-J.; Writing-creation of the original draft, M.M.; Writing-review and editing, M.M., H.P. and H.M.-J.; Visualisation, M.M.; Monitoring, M.M., H.P. and H.M.-J.; Project management, M.M., H.P. and H.M.-J.; Funding acquisition, M.M., H.P. and H.M.-J. All authors have read and agreed to the published version of the manuscript.

**Funding:** This research received no external funding.

**Institutional Review Board Statement:** Not applicable.

**Informed Consent Statement:** Not applicable.

**Data Availability Statement:** All required data in this article have only been published here and should be considered with respect to copyright.

**Conflicts of Interest:** The authors declare no conflict of interest.

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
