# Peer review of "Microstructural Properties of Particle-Reinforced Multilayer Systems of 316L and 430L Alloys on Gray Cast Iron"

_coatings, doi:10.3390/coatings13081450_

Round 1

Reviewer 1 Report

It is an interesting paper with well-discussed results and various findings. However, the following weakness should be modified before publishing in this journal.

1.      The abstract should be written better and needs major revisions. The purpose of research and innovation should be clearly stated. Also, the performed tests should be presented first, and then the results should be presented quantitatively and qualitatively.

2.      In the introduction section, the problem statement does not clearly explain why the authors did this research and what was missing in the previous studies that the authors addressed here:

Effect of whisker alignment on microstructure, mechanical and thermal properties of Mg-SiCw/Cu composite fabricated by a combination of casting and severe plastic deformation (SPD)

Effect of SiC particle size and severe deformation on mechanical properties and thermal conductivity of Cu/Al/Ni/SiC composite fabricated by ARB process

Influence of ARB technique on the microstructural, mechanical and fracture properties of the multilayered Al1050/Al5052 composite reinforced by SiC particles

Microstructure, Mechanical and Thermal Properties of Al/Cu/SiC Laminated Composites, Fabricated by the ARB and CARB Processes

Microstructure evolution and deformation behavior during stretching of a compositionally inhomogeneous TWIP-TRIP cantor-like alloy by laser powder deposition

3.      Some parts of the manuscript have poor English, so try to correct them.

4.      Please compare the results of this research with the published papers.

5.      Please shorten the conclusions. In the Conclusions, the chapter is better to add also the limitation and maybe future work.

6.      The results section is well organized and categorized. But some parts report the results, which require corrections and deepening the analysis and discussion.

Minor editing of English language required

Author Response

Replay to Review 1 for the article with the Nr. coatings-2498777

Dear Ladies and Gentlemen,

Thank you for your feedback and your time.

After careful observation of all the comments and feedbacks.

I would not like to join the details of feedback. After reading the articles mentioned in your review, I would like to say that they have nothing to do with the research project I am working on. For example, "Effect of whisker" SPD and ARB process" "CARB process" are not relevant to my project. Therefore, a comparison between my results and theirs is not correct and meaningful.

English of the text is again edited by a person who is doing a PhD in American Studies at the University of Marbug.

Nevertheless, the edited sections are highlighted in yellow sent it in to Ms. Linsuwanon.

 Once again, thank you for your feedback and I wish you all the best.

With kind regards

Mohammad Masafi

Reviewer 2 Report

Indeed, methods of surface engineering, in particular, laser technologies, have the right to life. They are effective and allow you to controllably change the microstructure and, accordingly, the functional properties. Therefore, the topic of the work is relevant. But there are significant shortcomings in the presentation of results.

 In particular

1. In the article, the authors did not explain why two-layer coatings of austenitic and ferritic steels were applied to the surface of gray cast iron using laser technology. The given argumentation is far from the truth. Taking this into account, the purpose of the work should be clarified and supplemented.

2. It raises doubts about the effectiveness of the chosen approach to applying the specified two-layer coating. For this, other more realistic and economically justified decision could be found.

3. In the photo (Figs. 2 and 3), the authors identify carbides of tungsten and/or titanium. How did they form in the studied steels with the specified chemical composition (Table 1). How the chemical affinity between alloying elements is taken into account during the laser alloying

4. The results of the chemical analysis of the selected steels do not agree with the SEM microscopy data (Fig. 7). These relate to both the identification of carbide phases and the sigma phase.

The article can be published after taking into account the specified comments.

Author Response

Replay to Review 2 for the article with the Nr. coatings-2498777

Dear Ladies and Gentlemen,

Thank you for your feedback and your time.

  1. The functions of each layer are described in detail.
  2. It is actually the aim of this project in the introduction to say the health and environment reasons are a strong reason for the economic story. On the other hand, according to the calculations, multilayer system must be used to achieve the target properties.
  3. It is also described in the text how the powder is mixed in advance in a container. Again, it can be observed that WC and TiC are already purchased and are not formed in our process.
  4. There is a thermodynamic phase reaction here. This means that the chemical composition of the microstructure does not match the chemical composition of the starting alloy.

English of the text is again edited by a person who is doing a PhD in American Studies at the University of Marburg.

The edited sections are highlighted in yellow sent it in to Ms. Linsuwanon.

Once again, thank you for your feedback and I wish you all the best.

With kind regards

Mohammad Masafi

Reviewer 3 Report

Dear authors:

In this work, the authors attempted to enhance the performance (e.g., wear and corrosion) of GJL through 316L/430L coating with hard particles by laser cladding. The multilayered microstructure was analyzed, especially at the interface, via materials characterization. After review, the manuscript is not considered appropriate for publication so far based on the following points. First, the background is not sufficiently introduced which leads to uncertain motivation. Second, the description of the experimental details and figures is not explicit. Third, the structure of manuscript is a bit disorganized, typically in section on results and discussion. Below are specific comments to be addressed and I hope the following comments will help improve this manuscript.  

Abstract/Introduction:

1. Line 15, the term “a Fe-based coating material” is confusing. Since this is abstract, I recommend that the authors can just say “using two types of stainless steel materials, namely 430L and 316L, with TiC and WC particles”. In addition, this work discussed multilayer systems. But the authors didn’t mention it in the abstract. Thus, I suggest the authors add one sentence or rephrase the abstract to highlight this point.

2. Lines 19-21, please remove duplicated sentence.

3. Lines 22-23, the sequence of keywords is not appropriate. “Gray cast Iron, Graphite phase, Laser Metal Deposition(LMD), Laser cladding, stainless steel coating, Sigma phase” could be better.

4. Lines 39-41, the statement “Therefore, the reduction of particle emissions and improvement of corrosion resistance with different coating processes have already been studied.” is confusing. Based on the introduction in lines 33-37, the particle emissions should be induced by wear/wear debris. Does corrosion also contribute to particulate emissions, and if so, how? Please introduce this important background. 

5. Lines 51-52, it looks like no difference between defects detection and coating characterization in the statement “the detection of possible defects and the characterization of coatings are urgently needed to improve and optimize the quality.” If the authors indicate the defects detection of the coating by some non-destructive techniques, that would be different from coating characterization. Please rephrase and clarify it.

Materials and methods

6. Line 58-Grammar, as shown/seen in Figure 1a. instead “see Figure 1a”. Please revise them in the manuscript.

7. Figure 1(b) and Figure 9, please clarify if these figures are adapted or reprinted from the cited source, or if the artwork is original and you are using only data from the source as follows.

8. Line 77, Please provide the final surface roughness level or detailed value. It is significant for wear and corrosion tests.

9. In Table 3, what is the thickness of each layer? Please provide it.

10. Please provide the information about the etching time.

11. Line 104, how did the authors conduct wear tests and analyze them? No information in the manuscript. Without evidence, how to draw a conclusion related to the increased or decreased wear performance?

Results and Discussion:

12. In Figure 3, I strongly suggest the authors add the original surface morphology as a comparison. The surface flake-off can be observed after the corrosion test, although (b) and (c) contains 316L layer. Does it indicate an adverse effect of hard particles on corrosion resistance? Which combination exhibits the best corrosion resistance?

13. In Figure 4, please clarify whether the cross section is after the cladding or after the corrosion test. If it is after the cladding, why did the authors organize it after Figure 3? If not, please also add the original cross section as a comparison and add some labels to clarify interfaces and “cracks start from the graphite” (the cracks seem to start from the top layer, rather than the substrate). In addition, the interface between GJL-316L in (b) is more damaged than the interface of GJL-316L in (c). Why do they have different levels of damage?

14. Is Figure 7 the surface or the cross section of the sample? It is confusing if Figure 7 shows the surface because the 430L has a low concentration of Ni. It would be difficult to observe Ni on the surface. Please specify and provide the thickness of the individual layers (comment 9).

Conclusions:

15. Please rephrase the statement in lines 174-176.

16. Lines 206-207, the conclusion is not convincing, based on the current observation and insufficient evidence.

17. The total number of references cited is 19, but in the manuscript, they are 18. Please check them carefully.

Moderate editing of English language required

Author Response

Replay to Review 3 for the article with the Nr. coatings-2498777

Dear Ladies and Gentlemen,

Thank you for your excellent feedback and your time.

In your review is my article very well read and understood and excellent commented.

Abstracts:

  1. Done
  2. Done
  3. Keywords are written as follows: from substrate to top layer and some new words are also added.
  4. Here it is better to correlate and compare the brake discs with and without coating and their influence on the production of dust emission.
  5. It is corrected although possible quality defects and their cause have been characterized. No new characterization method was invented here.

Materials:

  1. Done
  2. The pictures of the original surface have been added here.
  3. The roughness is added

  1. Thickness of each layer is added in Table 3.
  2. The etching time is given
  3. It was explained here like this:

"The focus of the microscopic examination was on analyzing wear, cracking, corrosion and Delamination. The corrosion test procedures and evaluation of the corrosion results are based on the test specification F.350.016 of the automotive industry and are not explicitly considered here."

But since a test according to customer standards is not allowed to be published here

Results:

  1. The picture of the original condition before the test is added and explained
  2. It is defined here that it is a cross-section and the text has been corrected as well
  3. Comments 9 is done, and it is also defined here that it is a cross-section. The Fig. 7 is observed and seen in the area of the top layer near each TIC.

Conclusions:

  1. Done
  2. The result and this summary have been edited and corrected.
  3. Done

English of the text is again edited by a person who is doing a PhD in American Studies at the University of Marburg.

The edited sections are highlighted in yellow sent it in to Ms. Linsuwanon.

Once again, thank you for your feedback and I wish you all the best.

With kind regards

Mohammad Masafi

Round 2

Reviewer 1 Report

Accept in present form.

Accept in present form.

Author Response

Reply to Reviewer 1 for Round 2 with the Nr. coatings-2498777(3)

Dear Ladies and Gentlemen,

Thank you again for accepting the manuscript.

I have improved the manuscript. The edited sections are highlighted in blue.

Minor editing of English language is done by a colleague fluent in English writing.

Once again, thank you for your feedback and I wish you all the best.

With kind regards

Mohammad Masafi

Reviewer 3 Report

Thank you for the authors’ work on revising the manuscript. I agree this is an interesting research work and the goal of this work has a practical significance, i.e., improving corrosion and wear resistance by multilayer coating.  However, after reviewing again, I regretfully maintain my previous decision after evaluating the entire manuscript, typically on the discussion of the relationship between microstructure and corrosion and wear.

1. The length of the introduction is almost more than 1500 words. Please remove inessential information and keep it concise.

2. I fully understand some testing details based on customer standards is not allowed to be published. But what is the shock corrosion test? Is it a salt spray corrosion test or an erosion test? I would suggest that the authors briefly explain it or use its other general name.

3. In terms of my previous comment 12, thank you the authors added the original surface morphology, but the authors didn’t demonstrate which original surface of combination is. If all three combination samples surface is similar, please clarify it in the manuscript. In line 275 and 276, the statement is ambiguous “In Figure 4c has been shown when the cracks arrive at the substrate (GJL), corrosion occurs more quickly due to the high electrochemical potential of GJL.” High electrochemical potential indicates noble in the corrosion thermodynamic. Why will noble GJL cause corrosion quickly? The corrosion that occurs more quickly may be due to galvanic corrosion between the substrate and 316L layer after cracks. In fact, galvanic corrosion is a very typical corrosion in multilayer systems. But in the manuscript, there is little discussion.

4.  Combined with my previous comment 11 and the authors' response, I agree with the microscopic examination of cracks, delamination, and even reluctantly agree with corrosion. However, I disagree with the microscopic examination on analyzing wear. It would be true that the top layer will have a high hardness due to the hard particles. But I might be skeptical about a microscopic observation on the wear resistance. In other words, high hardness does not equal high wear resistance, especially when the brittle phase is introduced. Therefore, the author's response does not convince me of their wear performance until evidence, such as wear volume, etc. is provided.

5. As suggested by the last reviewer, please shorten the conclusions.

Minor editing of English language required

Author Response

Reply to Reviewer 3 for Round 2 with the Nr. coatings-2498777(3)

Dear Ladies and Gentlemen,

Thank you again for your comments and review. I hope, this time this version will also convince you like other reviewers.

After the editing this is my pleasure to send you my manuscript with the required changes.

  1. Done in blue color.
  2. In line 189 to 192 is explained. The customer has named the test as shock corrosion test. This test complies with different DIN standards and in the meantime several hundred brakes were also done and these described in the text.
  3. here was given only picture of surface before the so-called corrosion test, since the all surfaces before the test look similar. I agree with your argument and will be gladly corrected and told a bit more: "The corrosion that occurs more quickly may be due to galvanic corrosion between the substrate and 316L layer after cracks. In fact, galvanic corrosion is a very typical corrosion in multilayer systems."

You find it in line 218-226:

In fact, galvanic corrosion is a very typical corrosion in multilayer systems. In Figure 4c has been shown when the cracks arrive at the substrate (GJL), the corrosion that occurs more quickly due to galvanic corrosion between the substrate GJL  with the high negative electrochemical potential of GJL (-0.6 until -0.8 V) and 316L layer (0.0 until  -0.2 V) after cracks. [https://link.springer.com/article/10.1007/s11837-005-0059-4]

galvanic beam speed is depend on the difference between the energy of active and noble.

The greater the difference, the faster corrosion is performed.

since the difference rate between galvanic energy 316L and GJL is greater than the difference between layer 430 L and layer 316L. therefore it can be reasoned that when the cracks reach GJL will corrode further faster.

  1. In one side is necessary for a good brake during high speed a high friction in short time, on the other side must be reduced dust emission.

I didn't mean that “high hardness does not equal high wear resistance, especially when the brittle phase is introduced” and I didn't put it in the text. I have written in line 167 to 169:

To reduce wear and emission particles, the second layer is coated on top of the first layer. Due to the hard particles, the second one can be used for wear reduction.”

The hard particles have very high hardness but matrix of coating has relative same hardness as substrate or bit more. Matrix of top layer coating helps to support and to hold the hard particles. While brakes generate enough friction and produce less dust emission. I will bring this small reasoning also in the text but of course I am also in the opinion, must be carried out as next steps for next article the further investigate and already planned [line 200 to 205]

  1. Done

Minor editing of English language is done by a colleague fluent in English writing.

The edited sections are highlighted in blue.

Once again, thank you for your feedback and I wish you all the best.

With kind regards

Mohammad Masafi
